# Dynamic inflow effects in measurements and high fidelity computations

Georg R. Pirrung[1] and Helge Aa. Madsen[1]

[1]Wind Energy Department, Technical University of Denmark, Frederiksborgvej 399, DK-4000 Roskilde, Denmark

*Correspondence to:* Georg R. Pirrung (gepir@dtu.dk)

**Abstract.** A wind turbine experiences an overshoot in loading after for example a collective step change in pitch angle. This overshoot occurs because the wind turbine wake does not immediately reach its new equilibrium, an effect usually referred to as dynamic inflow. Vortex cylinder models and actuator disc simulations predict that the time constants of this dynamic inflow effect should decrease significantly towards the blade tip. As part of the NASA Ames Phase VI experiment pitch steps have been performed on a turbine in controlled conditions in the wind tunnel. The measured aerodynamic forces from these experiments seemed to show much less radial dependency of the dynamic inflow time constants than expected when pitching towards low loading. Moreover the dynamic inflow effect seemed fundamentally different when pitching from low to high loading, and the reason for this behavior remained unclear in previous analyses of the experiment. High fidelity computational fluid dynamics and free wake vortex code computations yielded the same behavior as the experiments. In the present work these observations from the experiments and high fidelity computations are explained based on a simple vortex cylinder wake model.

## 1   Introduction

Models based on Blade Element Momentum (BEM) theory are commonly used in aeroelastic wind turbine codes. Blade element momentum theory describes how the forces acting on the wind slow down its velocity at the rotor disc, by superimposing a so-called induced velocity on the free wind speed. The induced velocity depends on the loading of the wind turbine rotor. When the rotor loading changes, for example due to a change in pitch angle, the induced velocity does not immediately reach a new equilibrium value, but instead slowly approaches it. Because BEM theory only predicts the steady state induction, dynamic inflow submodels are added to account for the delays in the induction response. The time constants of these models can be based on analytic derivations, results of higher fidelity models or measurements. Schepers (2007) compared analytically derived time constants for dynamic inflow with time constants derived from experimental data from the NASA Ames Phase VI measurements (Hand et al., 2001) and numerical results from the free wake vortex code AWSM (van Garrel, 2003). This comparison showed that the free wake code could predict the dynamic variations of the turbine loading measured in the wind tunnel with high accuracy. The analytical vortex cylinder model, on the other hand, predicted a strong dependency of the time constants on the radial position on the blade. Neither the experiment nor the free wake code results confirmed this predicted radial dependency. Investigating this disagreement was suggested as future research by Schepers (2012).

Sørensen and Madsen (2006) also investigated the dynamic inflow runs from the Phase VI experiment. They performed computational fluid dynamics (CFD) simulations using the EllipSys code (Sørensen, 1995). Similar to Schepers, they found that the observed dynamic inflow effects could be predicted by high fidelity codes, but the radial dependency of the time constants seemed much less pronounced than expected when deloading the rotor. Moreover the effect changed drastically when pitching from low to high loading, which is also in agreement with observations by Schepers. They suggested to use a dynamic inflow model with two time constants and scale the time constants with the wake velocity. The time constants have been obtained from actuator disc simulations.

More recently, Boorsma et al. (2016) compared results from the AWSM code with dynamic inflow measurements from the New Mexico experiment (Boorsma and Schepers, 2018). While the comparison is promising, the force overshoots are not as pronounced as in the Phase VI experiments due to the lower pitching rate relative to the rotor speed. A recent PhD thesis by Yu (2018) contains both experimental and numerical investigations of dynamic inflow effects. The experimental investigations use discs with varying porosity (Yu et al., 2017) and the numerical model consists of unsteady vortex rings for the near wake and a vortex cylinder model for the far wake (Yu et al., 2016).

In the following section, the NASA Ames experiment and the previous analyses by Schepers and Sørensen and Madsen are presented in more detail. Then a simple vortex cylinder model is introduced in Section 3. In Section 4 results from that model are used to explain the seemingly conflicting findings in previous work.

## 2  Previous analysis of NASA Ames experiments

In the NREL/NASA Ames Phase VI Experiments, Hand et al. (2001), a 2-bladed wind turbine with 10 m diameter has been placed in the NASA Ames open loop wind tunnel. One of the turbine blades was instrumented at 30%, 47%, 63%, 80% and 95% (corresponding to radii of 1.510 m, 2.343 m, 3.185 m 4.023 m and 4.780 m) with 22 pressure taps at each location. A large number of experiments in both parked and rotating conditions have been performed. The basis for the dynamic inflow investigations in Schepers (2007) are pitch step experiments at 5 $m/s$ wind tunnel speed and at a rotor speed of 72 rpm. In the relevant run, denoted as *Q0500000* in Hand et al. (2001), 20 pitch steps between -5.9 degrees pitch (heavily loaded rotor, induction factor $a \approx 0.5$) and 10.02 degrees pitch (unloaded rotor, $a \approx 0$) have been performed. The maximum pitch rate was roughly 66 degrees per second. This pitch rate corresponds to 6.6 deg/s for a turbine with 100 meter diameter, because the dynamic inflow time constants are inversely proportional to the rotor radius (Sørensen and Madsen, 2006). After each pitch step, a 15 second waiting time ensured that the flow conditions can reach an equilibrium. The previous analyses, as well as the present work, are based on the force response obtained by averaging the responses to the 20 pitch steps.

The measurement at 5 m/s exhibits prominent aerodynamic force overshoots that are characteristic for dynamic inflow. The dynamic response of the axial force after the upward and downward pitch steps is shown in Figure 1 in the left and right plot, respectively. Note that upward pitching refers to deloading of the rotor and downward pitching to loading of the rotor. This is based on the convention that the pitch angle is positive towards lower angles of attack. The force is scaled such that the

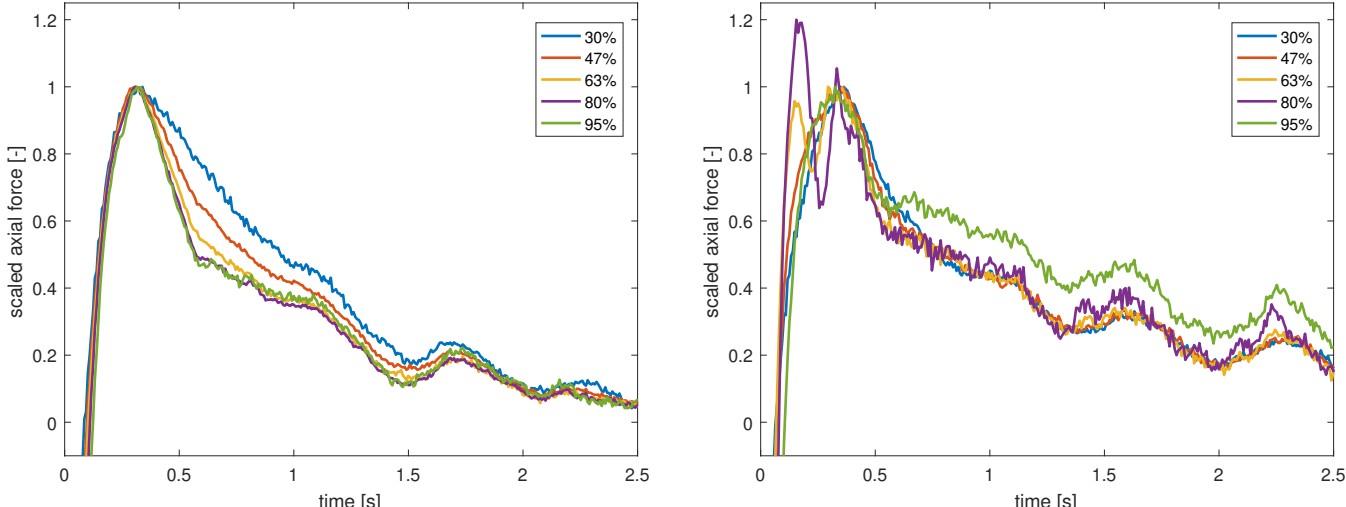

**Figure 1.** Scaled axial force measurements from the Phase VI experiment at 5 m/s. The pitch steps are deloading the rotor (induction factor from a=0.5 to a=0.0, left plot) and loading the rotor (a=0.0 to a=0.5, right plot).

overshoot after the pitch step is 1 and that the steady state force some time after the pitch step is 0:

$$F_{scaled} = (F - F_2)/(F_m - F_2), \tag{1}$$

where $F$ is the axial force, $F_2$ the equilibrium value of the force some time after the pitch step and $F_m$ the maximum or minimum force overshoot reached during the pitch step, depending on the pitching direction.

5      Some dynamics in the downward pitching step at the 80% radial station and to a much smaller extent at the 63% radial station appeared before the pitching has been finished at roughly 0.35 seconds. The maximum value at the 80% section has been increased to 1.2, see the right plot of Figure 1. This was not necessary for the 63% station because the maximum force overshoot at that station coincided with the remaining stations on the blade. The force response has been previously analyzed by Schepers (2007) and Sørensen and Madsen (2006). This previous work will be introduced in the following.

10   **2.1    Single time constant analysis by Schepers**

In the work by Schepers (2007) it was assumed that the behavior of the induction factor $a$ in time can be modeled using one first order low pass filter. For a step change in pitch the induction factor as a function of time is assumed to be:

$$a(t) = a_2 - (a_2 - a_1)e^{-(t-t_1)/\tau}, \tag{2}$$

where $a_1$ is the induction factor at $t_1$, before the pitch step, and $a_2$ is the equilibrium value of the induction some time after the 15   pitch step. The time constant is defined as $\tau$. It is then assumed that the forces are driven only by the change in axial induction factor, neglecting the faster 2D unsteady airfoil aerodynamics effects. This leads to an expression for the blade loads $F$:

$$F(t) = F_2 - (F_2 - F_1)e^{-(t-t_1)/\tau}, \tag{3}$$

| $r/R$ | $\tau_{analytic}$ | $\tau_{up,meas}$ | $\tau_{up,AWSM}$ | $\tau_{down,meas}$ | $\tau_{down,AWSM}$ |
|-------|-------------------|------------------|------------------|--------------------|--------------------|
| 0.30  | 0.93              | 0.95             | 1.12             | 1.07               | 1.98               |
| 0.47  | 0.83              | 0.83             | 1.04             | 1.09               | 1.61               |
| 0.63  | 0.68              | 0.77             | 1.01             | 1.10               | -                  |
| 0.80  | 0.44              | 0.74             | 1.00             | 1.13               | 1.14               |
| 0.95  | 0.14              | 0.78             | 1.03             | 1.53               | 2.36               |

**Table 1.** Time constants (in seconds) from measurements, AWSM calculations and cylindrical wake model, from Schepers (2007). The forces predicted by AWSM at the 63% radial station for the downward pitching step did not exhibit clear exponential behavior and therefore no time constant could be estimated.

where $F_1$, $F_2$ are the forces corresponding to the induction factors $a_1$ and $a_2$ defined above. The time constant $\tau$ can be determined as

$$\tau(t) = \frac{t - t_1}{\ln((F_2 - F(t))/(F_2 - F_1))}. \tag{4}$$

To avoid the initial overshoot and structural transients, this time constant was averaged between 0.7 and 3.7 seconds after the pitch step in Schepers (2007) for measurements, see Figure 1, and free wake code computations using the code AWSM (van Garrel, 2003). The computed time constants were then compared to the time constants obtained from a cylindrical wake model, see Section 3. This comparison is shown in Table 1. Comparing the time constants in the table leads to several observations:

1. For both upward and downward pitching case the time constants based on measurements and on free wake code computations vary much less with radial position than the analytically derived time constants predicted by the cylindrical wake model.

2. While the analytically derived time constants predict that the induction develops the fastest at the 95% section, this can't be seen in the measurements or AWSM computations. In the downward pitching case the tip time constants are even found to be largest at the blade tip.

3. The analytic time constants, which are based on the undisturbed inflow speed, are independent of the pitching direction, but there is considerable influence of the pitching direction on the dynamics seen in the measurements (Figure 1) and free wake computations.

## 2.2 Two time constant analysis by Sørensen and Madsen

The NASA Ames Phase VI dynamic inflow measurements were also analyzed by Sørensen and Madsen (2006). In their work the results were compared to full rotor CFD computations obtained from the EllipSys code (Sørensen, 1995) and BEM computations with a dynamic inflow model. The dynamic inflow model consists of two time constants, where the fast time

constant is radially varying and the slow time constant has much slower radial variation. These time constants were obtained from actuator disc simulations, and they are scaled with the wake velocity $v_w$:

$$\tilde{\tau} = \tau \frac{v_w}{R} \quad , \text{where} \quad v_w = v_\infty(1 - 1.5a). \tag{5}$$

In the above equation $v_\infty$ is the undisturbed inflow speed and $R$ is the wind turbine radius. The same scaling is applied to both the fast and the slow time constant.

5   The CFD computations agree well with the measurements showing a small radial variation of the time constants in general. In the CFD computations of the downward pitch step, loading the rotor, it was found that the tip forces seem to develop the slowest.

In a range of additional pitch step cases with a much smaller pitch angle variation computed by the CFD code the radial variation of the dynamic inflow time constants also appeared to be much less pronounced than in the actuator disc simulations.

## 3 Induction due to cylindrical wake

The engineering model that predicts the strong radial dependency of the time constants in Schepers (2007) is based on an analytical cylindrical vortex wake model. In this model, which has been presented by Snel and Schepers (1995), a semi-infinite cylindrical vortex is trailed from the tip radius at the rotor disc. Thus it is assumed that the rotor has an infinite number of blades with a radially constant bound circulation, and the root vortex is neglected. The induction from the trailed vortex cylinder is azimuth independent in the rotor plane but can vary radially. A sketch of the cylindrical wake is shown in Figure 2. The axial induced velocity $u_i$ at a radius $r$ on the disc due to such a cylindrical wake is (Snel and Schepers, 1995):

$$u_i = \frac{R}{4\pi} \int\limits_{x=0}^{x_{end}} \int\limits_{\phi=0}^{2\pi} \frac{\gamma_t(x)\left[R - r\cos\phi\right]}{\left[x^2 + R^2 + r^2 - 2rR\cos\phi\right]^{3/2}} d\phi dx, \tag{6}$$

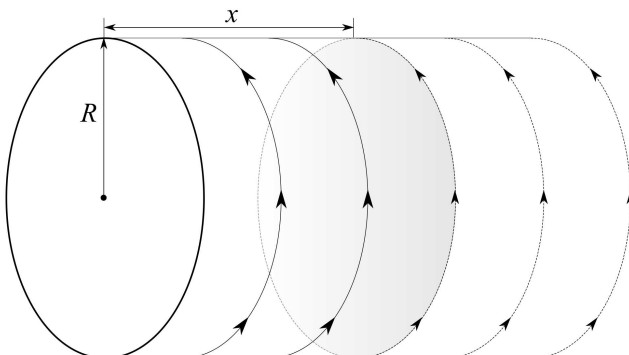

**Figure 2.** Sketch of cylindrical wake with radius $R$ and a step change in vortex strength at a distance $x$ from the rotor.

where $x$ is the stream wise coordinate, with $x_{end}$ denoting the length of the wake behind the rotor plane, and $\phi$ is the azimuthal coordinate. The rotor radius is denoted as $R$ and $\gamma_t$ is the tangential component of the wake vorticity, which induces an axial velocity in the rotor plane.

Previously this model was used to analytically derive the time constant for a step change in circulation at the rotor disc, which is transported downstream with the free stream velocity. In the present work, the position of the step change is varied and the wake velocity $v_w$, see Equation (5), is chosen as the downstream convection velocity. Equation (6) is integrated numerically.

## 3.1 Assumptions

The aim of the vortex cylinder computations presented here is to demonstrate that some of the fundamental questions from the NASA Ames Phase VI experiment and regarding dynamic inflow in general can be answered based on a very simple model. Therefore the following assumptions are made, neglecting some aerodynamic effects:

- The vortex strength of the cylindrical vortex wake changes from zero to a constant value at t=0.

  - 2D unsteady airfoil aerodynamics and dynamic stall are neglected. To justify this, the force response to a step change in angle of attack has been computed using Jones' equations for a flat plate and the relative speeds at the disc. After 0.35 seconds (corresponding roughly to the pitch step duration), the force response reaches between 90% (most inboard section) and 99% (tip section) of its quasi steady value, showing that this effect is much faster than the dynamic inflow effect.

  - pitching speed is neglected

- The downwind convection velocity of the trailed vortices is constant and equal to the wake velocity before the pitch step as given in Equation (5). Therefore the results at a time soon after the pitch step will be accurate, but the error grows over time.

- Wake expansion is neglected which introduces an error at the heavily loaded case

Further, in all plots the induced velocities $u_i$ at each radial station are scaled by the value $u_{i,\infty}$ they reach if the wake is 10 diameters long.

## 4 Results from cylindrical wake model

Figure 3 shows the induction as function of the dimensionless length of the cylindrical vortex wake behind the rotor. As written above, a constant downwind convection velocity is assumed in the computations. In the unloaded case ($a = 0$) this velocity is $v_w = 5m/s = 1R/s$. Therefore the dimensionless length of the wake in the unloaded case corresponds directly to the time, since the wake length after 1 second is 1 Radius (x/R=1). In the loaded case ($a = 0.5$) the wake velocity is 4 times smaller, see Equation (5), thus the non-dimensional wake length has to be multiplied by a factor of 4 to obtain the time.

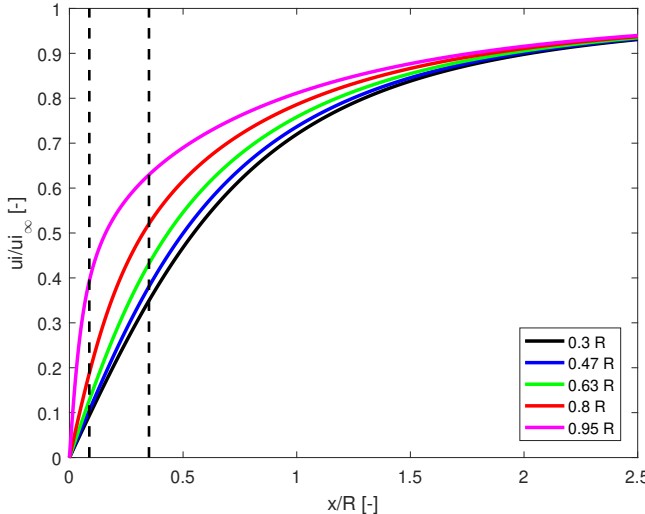

**Figure 3.** Normalized induced velocity at different radial stations depending on the length of the vortex wake behind the rotor plane. The dashed lines correspond to a time instant $t = 0.35s$ when the wake velocity is 1.25 m/s (left vertical line, corresponding to $a = 0.5$) and 5 m/s (right vertical line, corresponding to $a = 0.0$).

The induced velocity buildup clearly shows a large radial dependency at the beginning, but for a larger distance $x/R > 1.5$ of the vorticity step change from the rotor disc the radial dependency disappears. Then the dynamic induction buildup is similar on the whole disc.

## 4.1 Single time constant estimation

Based on Equations (2-4), the estimated time constant of a one term exponential function can be found as:

$$\tau(t) = \frac{t - t_1}{\ln((a_2 - a(t))/(a_2 - a_1))}. \tag{7}$$

Because the predicted time histories are not equal to one term exponential functions, the predicted time constants vary with time.

Applying Equation (7) on the induced velocities shown in Figure 3 leads to the estimated time constants as a function of the dimensionless length of the vortex wakes, which are shown in Figure 4. At $x_0/R \to 0$, the time constants predicted by the cylindrical wake model are equal to the analytically derived time constants provided by ECN, cf. Table 1. As the values of $x_0/R$ increase, the estimated time constants become less radially dependent, because the parts of the wake that are further away from the rotor have a more even influence on the whole rotor plane than the parts close to the rotor. These findings indicate, in agreement with the actuator disc computations presented by Sørensen and Madsen (2006), that a dynamic inflow model should employ two exponential terms. Then a small time constant can capture the radial dependency and a larger time constant captures the radially independent behavior on a longer time scale.

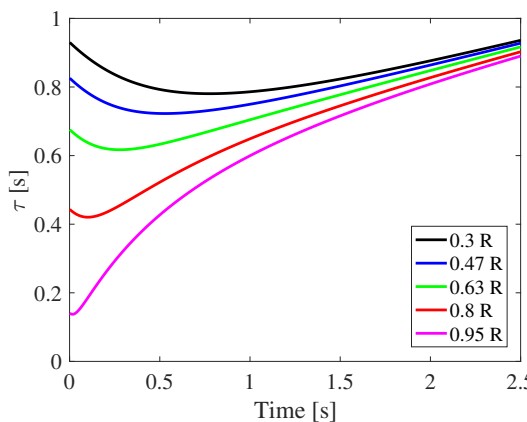

**Figure 4.** Estimated dynamic inflow time constants at different radial stations depending on the length of the vortex wake behind the rotor plane.

In Schepers (2007), the time constants obtained from the aerodynamic forces from measurements and free wake code results had to be averaged between $t = 0.7$s and $t = 3.7$s after the pitching step to avoid the initial force overshoot. The time constant estimation in Figure 4 shows how this method leads to moderate radial dependency, which explains some of the disagreements between analytical time constants and time constants estimated from measurements in Table 1.

## 4.2 Influence of pitching direction

Due to the assumption of a constant wake velocity and trailed vortex strength the different pitching directions only differ in how the wake length translates to the time after the pitch step, as explained at the beginning of Section 4. The force response decay in the Phase VI measurements can only be evaluated after the maximum of the force overshoot is reached, which is at roughly 0.35 seconds, see Figure 1. These 0.35 seconds correspond to a non-dimensional wake length of either 0.35 (pitching from unloaded to loaded conditions) or 0.0875 (pitching from loaded to unloaded conditions), as indicated by the dashed lines in Figure 3. To see what influence this delayed evaluation has, the induced velocity has been rescaled such that the value at 0.35 seconds corresponds to 0, and the value at steady state to 1. These rescaled induced velocities are shown in Figure 5. For the pitch step starting in loaded conditions (left plot) a clear radial dependency of the induced velocities is seen, which develops slowly inboard and faster outboard. In the right plot, where the rotor is not loaded before the pitch step, the induced velocity seems to develop similarly fast for all sections except the tip sections, where the development is slowest. A comparison with the scaled measured force response in Figure 1 shows the same influence of the pitching direction. This indicates that the dynamic inflow effect is mainly governed by the downstream convection speed of the tip vortices. Further, the decreased or reversed radial dependency in the force measurements from the NASA Ames Phase VI experiment does not contradict the radial dependency in the induced velocity time constants that is expected from previous investigations using vortex cylinder models and actuator discs. Instead, the apparent reversal of the radial time constant dependency with pitching direction is observed because the decay of the dynamic forces can only be analysed some time after the pitch step is completed. At this

time, the induced velocity has already developed considerably, which has a large influence on the size of the overshoot. This initial induced velocity development is missed when estimating time constants on the scaled time history of the forces.

This conclusion holds also if the force time series are obtained from CFD simulations. Another option is to use dynamic inductions from the CFD solution to estimate the time constants. However, a recent comparison has shown that there is still considerable uncertainty on induced velocities from CFD towards the root and tip of the blade (Rahimi et al., 2018). This uncertainty complicates the estimation of the radially varying dynamic inflow time constants. Jost et al. (2018) state the following on the comparison between BEM based models and CFD: '*While the approach for loads with respect to the rotor-plane coordinate system such as distributed driving force or thrust is straightforward, a comparison of aerodynamic characteristics is more challenging*'. They present a promising methodology for extracting the unsteady induced velocities that can capture the increasing induction towards the blade tip.

The induced velocities at the blades or at an actuator disc are directly available in vortex codes. Yu (2018) has developed an engineering dynamic inflow model based on results from a combined unsteady vortex ring and vortex cylinder model. The behavior of the unsteady cases investigated in Yu's thesis was such that the increasing load case reacted slower than the decreasing load case. This appears to contradict the results from the Phase VI measurements at first glance, but the cases were fundamentally different: the increasing load case is defined as moving from middle to high loading, the decreasing load case from middle to low loading. Thus the average wake velocity is lower in the increasing load case, leading to a slower dynamic inflow response. In the Phase VI experiments, the pitch steps change the load from low to high and high to low, respectively. Then the wake velocity before the pitching step governs the immediate dynamic inflow response and the case with increasing load responds faster. Also, Yu was able to study the effects of a roll-up of the vortex rings which would not be possible with the simple model used in the present work. This shows the importance of using higher fidelity models for more detailed investigations.

## 5   Conclusions

A simple vortex wake model has been used to investigate the question why neither the unsteady NASA Ames Phase VI experiments nor the corresponding high fidelity simulations showed the expected radial dependency of the dynamic inflow time constants. Results from the model show that the dynamic inflow response initially exhibits a strong radial dependency, but a short time after the step change in loading, the time constants for the different stations become similar, suggesting that two time constants are necessary for accurate modeling of dynamic inflow effects. When using the force response as a basis for time constant investigations, time constants can only be analyzed shortly after the pitch step when the forces start decaying from the maximum overshoot. This delayed analysis obscures some of the fast induction development and can lead to an apparent decreased or even reversed radial dependency.

This indicates that it is difficult to obtain time constants for dynamic inflow models based only on the force time series from measurements, CFD or free wake vortex code simulations of a wind turbine. These force measurements and computations though have a large value in validating the aerodynamic force predictions from engineering models. Computing unsteady

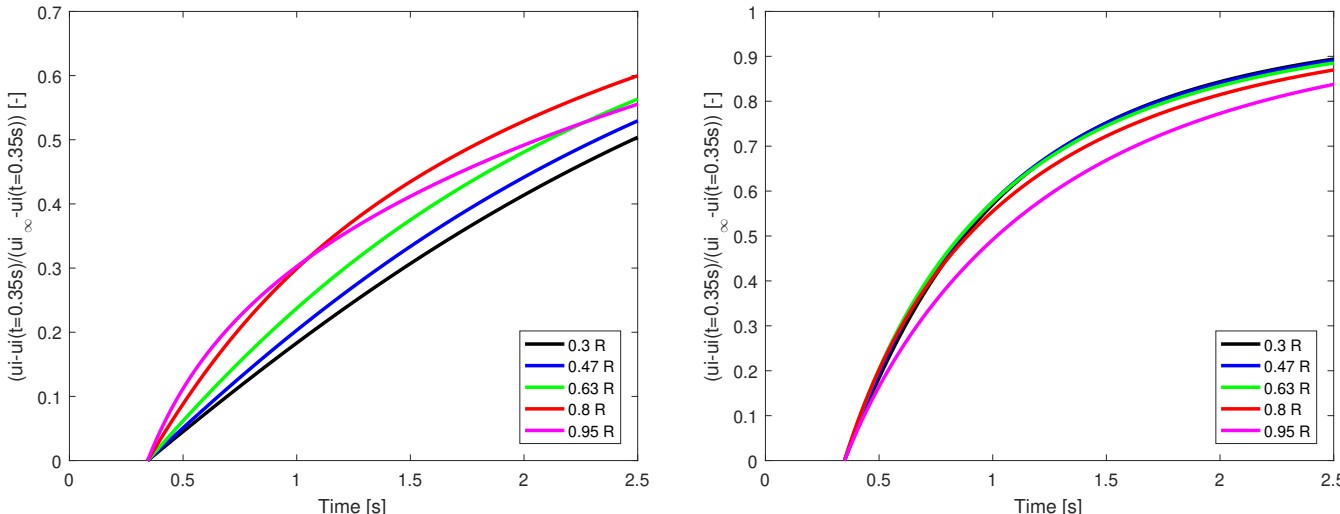

**Figure 5.** Scaled induced velocities predicted by the cylindrical wake model for the deloading (left plot) and loading (right plot) of the rotor. The scaling is such that the velocities range from 0 at the time instant just after the maximum force overshoot (indicated by dashed vertical lines in Figure 4) to 1 . The scaled induced velocities show the same qualitative trends as observed in the force measurements, see Figure 1.

induced velocities from CFD simulations is an active field of research and there is still some uncertainty, especially towards the blade root and tip. Free wake vortex codes, on the other hand, could be used more easily to investigate dynamic inflow effects and to tune engineering models, because the induced velocity is directly available from the computations.

*Acknowledgements.* The work has been carried out within the project "Dansk deltagelse i IEA Wind Task 29 Mexnext III" granted by the
5   Danish funding agency EUDP, grant J.nr. 64014-0543.

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
