# Peer review of "Dynamic inflow effects in measurements and high fidelity computations"

_Wind Energy Science, 2018_

## Referee Comment (RC1) · Anonymous Referee #1 · 23 Mar 2018

The article deals with the so-called dynamic inflow effect, which takes place after a step change in pitch angle (among other situations involving a fast change of the rotor loading). The authors address some issues that had been described by other authors in the past and for which no satisfactory explanation exists so far. More specifically, the article focuses on the radial dependency of the dynamic inflow constants when pitching both towards lower and higher loading. Experimental and numerical observations done by other authors on the NREL Phase VI wind turbine are discussed in detail and explanations for the mentioned effects are presented. For this, the authors rely on the results obtained by means of a simple vortex cylinder wake model. The topic of the article is of sufficient relevance for the wind energy community and the questions addressed are within the scope of WES. The objectives of the article are clear. The scientific method

employed is appropriate for the current study. The discussion, as fas as this referee can judge, is relevant and backed up. The paper is well structured and well written. The authors give proper credit to relevant work and the literature is correctly cited. The following issues, however, should be addressed before the paper can be accepted for publication in this journal.

MAJOR COMMENTS:

1- The method of your analysis has already been applied by other authors for the study of the same effect on the same turbine. Your interpretation of the results is the main novelty of the paper since it contributes to understanding the pros and cons of the methods used in the past for the analysis of this aerodynamic effect. However, my impression is that no new knowledge about the dynamic inflow effect itself is generated with this work. This referee expects a more clear description of what we learn from this work with regard to the dynamic inflow effect (not with respect to its modeling).

2- The authors conclude that the use of a simple vortex wake model is preferred over other methods for studying dynamic wake effects because the induction in the rotor can be obtained from the model, what allows to obtain the time constants for dynamic inflow models without the detour of using aerodynamic forces. The vortex wake model, however, is based on rather crude assumptions, which are also described in the manuscript. Therefore, this referee would expect that obtaining the wake induction from e.g. CFD results and applying it to eq. 6 would be a more accurate and reliable method for determining the mentioned time constants.

3- A discussion on the applicability of the current results to large wind turbines is absolutely required. The pitch rate of the NREL Phase VI wind turbine was 66 degrees per second. This is very far from a realistic pitch rate for modern wind turbines. Which is, therefore, the relevance of this study for practical applications?

MINOR COMMENTS:
1- Is the vortex wake model the same as the one applied by Schepers (2007)? Is it a different implementation? What is new or different in your model?

2- Page 2, line 28: the load fluctuations are not only to be seen at the 80% station but also, to a lower extent, at the 63% radial station.

3- Page 2, last paragraph. Please state when the pitching step was finished.

4- Page 3, line 8: how realistic is to neglect 2D unsteady airfoil effects? Do you expect this assumption to have an impact on the reliability of your results?

5- Eq. 2: is the relationship between the induction and the loads always linear? What would happen if a low induction leads to the onset of stall? Can you still, in that case, apply this equation?

6- You refer several times to the simple vortex wake model as the analytical model and its results are also termed as analytical. Please explain why you consider this model to be analytical in contrast to the other models that have been used by other authors for the study of this effect.

7- Page 5, 1st paragraph: do you use any type of tip correction and why?

8- Page 5, line 15: is the flow non-uniformity of the rotor plane also neglected?

9- Page 6, line 10: please explain more clearly why in the loaded case the wake velocity is 4 times smaller than in the unloaded case.

10- Some minor typos exist throughout the whole text. Please correct them. Examples:

page 9, line 10: off → of

page 2, line 18: performet → performed

---

## Referee Comment (RC2) · Anonymous Referee #2 · 24 Mar 2018

General Comments: The authors use an interesting and simple analytical model to examine conflicting findings from previous experimental and computational work regarding dynamic inflow effects. The paper is well written and concise and I believe contribute provides useful information that should be shared with the wind energy community.

Specific Comments: 1. On line 26 of pg2, "The force is scaled such that..." I think it would be helpful to the reader to know more explicitly how this scaling of the force was done. I would suggest either labeling the ordinate axis in Figure 1 with this scaling (rather than just "scaled axial force") or including it as an equation in the text.

2. In equation 4 on pg4, it is not clear from the text which Tau is the fast time constant and which is the slow.

[Figure]

3. In table 1 of pg4, why is there no entry for the Tau_down AWSM calculation?

4. I understand that the analytical model presented neglects unsteady aerodynamics & dynamic stall, but can the authors comment on how these unsteady effects might affect the asymmetrical force response between pitch up and pitch down maneuvers?

Technical corrections: 1. Typographical error in the conclusion: "off course" should be "of course"

---

## Author Comment (AC1) · 13 Apr 2018

Dear reviewer,

Thank you very much for the detailed review. We have clarified some points in the article and hope that we have sufficiently answered your questions below (the answers are indented).

MAJOR COMMENTS:
1- The method of your analysis has already been applied by other authors for the study of the same effect on the same turbine. Your interpretation of the results is the main novelty of the paper since it contributes to understanding the pros and cons of the methods used in the past for the analysis of this aerodynamic effect. However, my impression is that no new knowledge about the dynamic inflow effect itself is generated with this work. This referee expects a more clear description of what we learn from this work with regard to the dynamic inflow effect (not with respect to its modeling).

- It seems difficult to differentiate clearly between what we learn with respect to the effect from what we learn about modeling of the effect. The main points are that the apparently conflicting findings from previous work between experiment, high fidelity models and simple models are actually in agreement. But they appeared conflicting because the dynamic inflow analyses from measurements and high fidelity had to be based on force responses. This work suggest that the dynamic induction change predicted by a simple vortex model does not - as thought before - contradict the measurements and high fidelity computations. This shows that the effect itself is less complex than previously thought.

2- The authors conclude that the use of a simple vortex wake model is preferred over other methods for studying dynamic wake effects because the induction in the rotor can be obtained from the model, what allows to obtain the time constants for dynamic inflow models without the detour of using aerodynamic forces. The vortex wake model, however, is based on rather crude assumptions, which are also described in the manuscript. Therefore, this referee would expect that obtaining the wake induction from e.g. CFD results and applying it to eq. 6 would be a more accurate and reliable method for determining the mentioned time constants.

- There seems to be a misunderstanding. Definitely CFD would be preferred over the presented simple vortex wake model when computing the force time series. The purpose of this article, however, is to show that it is difficult to extract dynamic inflow time constants from time series of the force response. It is also quite difficult to obtain a reliable unsteady induced velocity from CFD, because the induced velocity has to be computed from the flow field. This is why we state in the conclusions that free wake vortex codes (that are much more complex than the model used in the present work) have an advantage over CFD computations when it comes to determining dynamic inflow time constants: the time series of the induced velocity at the blades is directly available from the free wake vortex computations.

3- A discussion on the applicability of the current results to large wind turbines is absolutely required. The pitch rate of the NREL Phase VI wind turbine was 66 degrees per

second. This is very far from a realistic pitch rate for modern wind turbines. Which is, therefore, the relevance of this study for practical applications?

- Dynamic inflow time constants are inversely proportional to the rotor radius, see Equation (4) in the article. Therefore, at the same free wind speed, dynamic inflow effects are 10 times slower for a moderately sized modern turbine with a diameter of 100 meters than for the Phase VI turbine (10 meters diameter). Thus, for the same ratio between pitch rate and dynamic inflow time constant a pitch rate of 6.6 degrees/second is sufficient. Since the step size of 16 degrees used in the experiment is very large, the pitch rate can be even lower if steps of a smaller size are considered. Thus the effects discussed can definitely occur on a modern turbine.
- The following sentence has been added: '*This pitch rate corresponds to 6.6 deg/s for a turbine with 100 meter diameter, because the dynamic inflow time constants are inversely proportional to the rotor radius (Sørensen and Madsen, 2006).'*

MINOR COMMENTS:
1- Is the vortex wake model the same as the one applied by Schepers (2007)? Is it a different implementation? What is new or different in your model?

- The following clarification has been added in the last paragraph before section 3.1:

  *Previously this model was used to analytically derive the time constant for a step change in circulation at the rotor disc, which is transported downstream with the free stream velocity. In the present work, the position of the step change is varied and the wake velocity v_w, see Equation (4), is chosen as the downstream convection velocity. Equation (5) is integrated numerically.*

2- Page 2, line 28: the load fluctuations are not only to be seen at the 80% station but also, to a lower extent, at the 63% radial station.

- This is correct, and we state it in the revised article.

3- Page 2, last paragraph. Please state when the pitching step was finished.

- At roughly 0.35 seconds in the plot where the maximum force overshoots are observed (except at the 80% station). This is stated in the revised article.

4- Page 3, line 8: how realistic is to neglect 2D unsteady airfoil effects? Do you expect this assumption to have an impact on the reliability of your results?

[Figure]

- at 0.35 seconds, the value of the most inboard section has reached roughly 90% of the final value, the most outboard section 99%. This is much faster than the dynamic inflow effect (see Figure 3 in the article). There is no strong influence of the induction because the time constants depend on the local relative speeds, which are dominated by the relative speed due to rotor rotation. Therefore they don't add any more asymmetry between the loading and deloading of the rotor.

The following has been included in the article: *2D unsteady airfoil aerodynamics and dynamic stall are neglected. To justify this, the force response to a step change in angle of attack has been computed using Jones' equations for a flat plate and the relative speeds at the disc. After 0.35 seconds (corresponding roughly to the pitch step duration), the force response reaches between 90\% (most inboard section) and 99\% (tip section) of its quasi steady value, showing that this effect is much faster than the dynamic inflow effect.*

5- Eq. 2: is the relationship between the induction and the loads always linear? What would happen if a low induction leads to the onset of stall? Can you still, in that case, apply this equation?

- No, this is an assumption that fails for stalled areas of the blade. Simulations by Schepers showed however that stall only occurs immediately after the Pitch step in the case towards higher loading (see Figures 26 and 27 in Schepers (2007) and the flow attaches within 2 seconds. While this introduces some uncertainty, the averaging of the estimated

time constants between 0.7 to 3.7 seconds will lead to improved results here, since the flow is attached for most of this window.

6- You refer several times to the simple vortex wake model as the analytical model and its results are also termed as analytical. Please explain why you consider this model to be analytical in contrast to the other models that have been used by other authors for the study of this effect.

- Analytical is used to state that the time constants can be derived directly from the model as opposed to using curve fitting on other models. This term has also been used by Schepers (2007). It doesn't mean in any way that the model is superior to other models.

7- Page 5, 1st paragraph: do you use any type of tip correction and why?

- No tip correction is used. We have investigated the same effects using a helical vortex model (where the single blades are represented as opposed to the disc approach in the article). The conclusions on the radial dependency and the influence of the convection velocity from that model have been identical, so no tip loss correction is needed to show this behavior. It appears that the cylindrical wake model is the simplest model that is sufficient to explain the dynamic inflow effects. Therefore we focused on this model and removed the slightly more complex helical model to make the article more clear and concise.

8- Page 5, line 15: is the flow non-uniformity of the rotor plane also neglected?

The following sentence has been added in Section 4:
*'The induction from the trailed vortex cylinder is azimuth independent in the rotor plane but can vary radially.'*

9- Page 6, line 10: please explain more clearly why in the loaded case the wake velocity is 4 times smaller than in the unloaded case.

- We added the definition of the wake velocity in Equation 5. We clarified the explanation by adding a reference to Equation 5.

10- Some minor typos exist throughout the whole text. Please correct them. Examples:

page 9, line 10: off → of

- this is corrected

page 2, line 18: performet → performed

- this is corrected

---

## Author Comment (AC2) · 13 Apr 2018

Dear reviewer,

Thank you very much for the review. Please find our answers below (indented). A brief clarification to each point has been added in the article.

On line 26 of pg2, "The force is scaled such that. . ." I think it would be helpful to the reader to know more explicitly how this scaling of the force was done. I would suggest either labeling the ordinate axis in Figure 1 with this scaling (rather than just "scaled axial force") or including it as an equation in the text.

- This Equation has been added (now Equation (1)).

In equation 4 on pg4, it is not clear from the text which Tau is the fast time constantand which is the slow.

- The following sentence has been added: *'The same scaling is applied to both the fast and the slow time constant.'*

In table 1 of pg4, why is there no entry for the Tau_down AWSM calculation?

The time series did not exhibit clear exponential behavior; *'therefore no time constant could be estimated'* has been added in the caption of table 1.

I understand that the analytical model presented neglects unsteady aerodynamics& dynamic stall, but can the authors comment on how these unsteady effects might affect the asymmetrical force response between pitch up and pitch down maneuvers?

- see answer to question 4, reviewer 1

Technical corrections:

Typographical error in the conclusion: "off course" should be "of course"

- This has been corrected in the revised article.

---

## Author Response (AR2)

Dear editor,

You wrote:'*The manuscript has been positively reviewed. There are some outstanding points that have not been addressed extensively in the manuscript concerning reviewer #1 (see major points one and two). The results section could be expanded to reflect these.*'

To better address these points, the results section and conclusion section have been extended:

1) It is highlighted that the important mechanism governing the dynamic inflow effect is the tip vortex convection velocity (major point 1)
2) It is explained in more detail that the limitations of the CFD methods for determining dynamic inflow time constants are not due to the modeling accuracy, which is far above the simple model used in the article. They are due to the difficulties encountered when extracting dynamic induced velocities from the CFD solutions, especially close to the root and tip of the blade. (major point 2). A reference on a recent comparison of methods for extracting angle of attack and induced velocity has been added to support this statement.

Please find the detailed changes in the results and conclusions section below.

[revised manuscript text omitted]

---

## Author Response (AR3)

Dear editor,

You wrote: *'The manuscript should further reflect and embed the comments as provided by the reviewers'*

To better address these points, the introduction and the results section have been expanded:

1) We added a paragraph on recent research (both numerical and experimental) concerning dynamic inflow in the introduction
2) It is made more clear that the time constants of the induced velocity can indeed be radially dependent, even if the time constants of observed force measurements or force computations appear independent of the radial position. This removes some previous uncertainty about the mechanism behind the dynamic inflow effect.
3) We extended the paragraph concerning extraction of induced velocities from CFD and refer to some very promising recently published work.
4) We explain how the findings from a recently published PhD thesis are related to the present article.
5) Six additional references have been added.

Please find the changes in the 'diff' below.

[revised manuscript text omitted]